# Use of Microalgae-Derived Astaxanthin to Improve Cytoprotective Capacity in the Ileum of Heat-Induced Oxidative Stressed Broilers

**DOI:** 10.3390/ani14131932

**Published:** 2024-06-29

**Authors:** Donna Lee Kuehu, Yuanyuan Fu, Masaki Nasu, Hua Yang, Vedbar S. Khadka, Youping Deng

**Affiliations:** 1Bioinformatics Core, Department of Quantitative Health Sciences, John A Burns School of Medicine, University of Hawaii, Honolulu, HI 96813, USA; dkuehu@hawaii.edu (D.L.K.); fuy@hawaii.edu (Y.F.); mnasu@hawaii.edu (M.N.); yanghua@hawaii.edu (H.Y.); vedbar@hawaii.edu (V.S.K.); 2Department of Molecular Biosciences and Bioengineering, College of Tropical Agriculture and Human Resources, University of Hawaii at Manoa, Honolulu, HI 96822, USA

**Keywords:** astaxanthin, cytoprotective capacity, ileum, heat stress, oxidative stress, broilers

## Abstract

**Simple Summary:**

Oxidative stress exerts profound effects on intestinal ileum tissue, contributing to the pathogenesis of gastrointestinal disorders, including inflammatory bowel disease, intestinal ischemia, and intestinal permeability dysfunction. The delicate balance between reactive oxygen species and antioxidant defenses in the ileum is crucial for maintaining cellular homeostasis and function. In this research, the aim was to maintain growth performance indicators such as weight gain, daily feed intake, and feed conversion efficiency, and to protect ileum epithelial integrity by increasing the cytoprotective capacity using astaxanthin antioxidants. Using heat as an inducer of oxidative stress, we compared the effects on the ileum of broilers for gene expression and tissue morphology between three groups, i.e., thermal neutral, heat stress, and heat stress with astaxanthin. Results showed that the thermal neutral group had better growth performance indicators over the two groups treated with heat; however, the heat stress with the astaxanthin group showed a slightly lesser decline. In addition, the heat stress with astaxanthin treatment group was upregulated in the cytoprotective and epithelial integrity genes. Research on the effectiveness of antioxidant feed additives holds promise as mitigation strategies to maintain poultry health from heat-induced oxidative stress and damage by improving cytoprotective capacity and epithelial integrity in the ileum.

**Abstract:**

The gastrointestinal tract has a pivotal role in nutrient absorption, immune function, and overall homeostasis. The ileum segment of the small intestine plays respective roles in nutrient breakdown and absorption. The purpose of this study was to investigate the impact of heat-induced oxidative stress and the potential mitigating effects of an astaxanthin antioxidant treatment on the ileum of broilers. By comparing the growth performance and gene expression profiles among three groups—thermal neutral, heat stress, and heat stress with astaxanthin—thermal neutral temperature conditions of 21–22 °C and heat stress temperature of 32–35 °C, this research aims to elucidate the role of astaxanthin in supporting homeostasis and cellular protection in the ileum. Results showed both treatments under heat stress experienced reduced growth performance, while the group treated with astaxanthin showed a slightly lesser decline. Results further showed the astaxanthin treatment group significantly upregulated in the cytoprotective gene expression for *HSF2, SOD2, GPX3*, and *TXN*, as well as the upregulation of epithelial integrity genes *LOX, CLDN1*, and *MUC2*. In conclusion, our experimental findings demonstrate upregulation of cytoprotective and epithelial integrity genes, suggesting astaxanthin may effectively enhance the cellular response to heat stress to mitigate oxidative damage and contribute to cytoprotective capacity.

## 1. Introduction

The gastrointestinal tract (GIT) is essential for digesting food into basic components for optimal nutrient absorption, crucial for poultry production and welfare. Bacteria in the GIT protect against disease, stimulate the immune system, and enhance growth rates by producing extra nutrients through fermentation of non-digestible fibers [1]. The chicken GIT consists of five regions, each with a specific role, i.e., the crop, proventriculus, gizzard, small intestine (duodenum, jejunum, and ileum), and large intestine (ceca, colon, and rectum). In the crop, food is partially fermented before entering the proventriculus, and acid and enzymes aid in protein breakdown. The gizzard grinds food into smaller particles for the small intestine, where most nutrient absorption occurs, leaving behind non-digestible components. The ceca further ferments these components, forming organic acids, short-chain fatty acids, and vitamins for absorption [2,3]. The small intestine, the primary site for nutrient absorption, is divided into the duodenum, jejunum, and ileum. Digestive enzymes and bile aid nutrient breakdown, and villi increase absorption. The ileum absorbs remaining nutrients and plays a role in immune function via Peyer’s patches [1,4]. Impaired GIT function leads to reduced nutrient absorption, poor feed conversion efficiency, and higher feed costs. Dysbacteriosis, an imbalance of gut bacteria, negatively affects chickens by reducing nutrient absorption, often causing increased feed consumption to meet nutritional demands [1,5,6].

The ileum, the final section of the small intestine, is crucial for nutrient absorption before passing digested material to the ceca. Its wall is lined with villi, tiny projections that increase the surface area for absorption. Villi contain capillaries that transport amino acids and glucose to the hepatic portal vein and liver [7,8,9]. The lysyl oxide (*LOX*) gene encodes lysyl oxidase, an enzyme essential for crosslinking collagen and elastin fibers in the extracellular matrix (ECM), vital for connective tissue integrity and stability. Although not primarily expressed in epithelium, *LOX* activity is important in epithelial function, particularly under oxidative stress, aiding in wound healing and tissue repair by reorganizing and strengthening the ECM [10,11]. Claudins (*CLDN1*) are integral membrane proteins fundamental to forming and regulating tight junctions (TJs) in epithelial and epithelial cell barriers. TJs control the passage of ions, water, and solutes across cells, maintaining barrier integrity. Changes in *CLDN1* expression can affect tight junction permeability, with certain isoforms enhancing or compromising barrier function [12]. The ileal mucosa protects and maintains epithelial integrity, partly through the mucin 2 *(MUC2*) gene, which synthesizes mucin 2 glycoprotein to maintain the mucus layer, protecting cells from oxidative damage. A disruption of these factors can increase epithelial permeability. The mucosa regulates gut inflammation, producing anti-inflammatory factors and specialized mediators to restore and maintain epithelial integrity. It interacts with gut microbiota to balance beneficial and harmful microbes, supporting a healthy mucosa. Mucosal immune responses defend against pathogens and support beneficial bacteria, contributing to a healthy and functional barrier in the small intestine [13,14].

Heat-induced oxidative stress profoundly affects intestinal ileum tissue and contributes to gastrointestinal disorders such as inflammatory bowel disease, intestinal ischemia, and intestinal permeability dysfunction [15,16,17]. The balance between reactive oxygen species (ROS) and antioxidant defenses in the ileum is crucial for cellular homeostasis and function. Enhancing cytoprotective capacity with the use of antioxidants or induction of heat shock proteins can protect cells from harmful stimuli [18,19,20]. Cytoprotective capacity refers to the ability to shield cells from damage or death caused by various stressors, including oxidative stress, toxins, radiation, and inflammation [21,22]. Antioxidants (AOXs), such as superoxide dismutase (*SOD*), catalase (*CAT)*, and glutathione peroxidase (*GPX*), neutralize ROS, but their effectiveness can diminish under elevated temperatures, increasing ROS and oxidative stress [23,24,25]. The thioredoxin (*TXN*) gene encodes thioredoxin proteins (Trx), which play a vital role in cellular redox homeostasis and oxidative stress defense by maintaining a reducing environment through thiol-disulfide exchange reactions. Under oxidative stress, *TXN* activity is upregulated to enhance antioxidant defenses, protecting cells from ROS damage [6]. Oxidative stress in the gastrointestinal system can compromise intestinal mucosa integrity, disrupt cellular function, and cause inflammation. Understanding heat-induced oxidative stress mechanisms in the small intestine is essential for developing strategies to protect intestinal health [7]. Heat shock proteins, produced under stress, assist in protein folding and function, protecting cells from damage. Heat shock response, activated by elevated temperatures, involves heat shock factor 2 (*HSF2*) binding to heat shock elements in gene promoters. HSF2 helps manage protein misfolding and aggregation, maintaining cellular homeostasis and preventing cell damage or death [26,27,28].

Astaxanthin (AST), a potent lipid-soluble antioxidant from the carotenoid family, exhibits remarkable free radical scavenging and anti-inflammatory properties [29,30]. Its unique chemical structure, with hydroxyl and keto moieties on each ionone ring, contributes to its superior antioxidant activity, protecting cellular membranes against oxidation [31]. Although naturally occurring in aquatic organisms and birds, astaxanthin cannot be synthesized by animals and must be obtained from the diet [32]. Common natural sources include green algae, red yeast, and crustacean byproducts, with the highest concentration found in *Haematococcus pluvialis* [33]. The carotenoid antioxidant value of powder astaxanthin is described as the Oxygen Radical Absorbance Capacity (ORAC) and expressed in micromoles of Trolox equivalents per 100 g (μmol TE/100 g), a vitamin E analog. Astaxanthin has been shown to have 6000 times the antioxidant capacity of vitamin C, 800 times that of CoQ10, and 550 times that of vitamin E [34]. Astaxanthin supplementation in humans has shown positive health benefits, including anti-inflammatory, immunomodulatory, cardiovascular, neuroprotective, and anticancer effects [30].

Our research question is, “Does astaxanthin reduce oxidative stress in the broiler ileum, and what genes are responsible for regulating its cytoprotective capacity and epithelial integrity?” We hypothesize that heat stress induces oxidative stress in the broiler ileum, correlating with poor growth performance and tissue structure degradation, and *H. pluvialis*-derived astaxanthin dietary supplementation mitigates the effects through upregulation of cytoprotective and epithelial integrity gene expression.

## 2. Materials and Methods

### 2.1. Ethics Statement

Animal protocol (Protocol No. 17-2605) used in this study was approved by the University of Hawaii Institutional Animal Care and Use Committee (IACUC). Animals were raised under animal welfare guidelines and euthanized in accordance with humane protocols in preparation for necropsy.

### 2.2. Experimental Animal Design

Cobb 500 unsexed broiler chicks were obtained from Asagi Hatchery (Honolulu, HI, USA). Asagi Hatchery is a local commercial producer that sells newly hatched chicks for commercial or research purposes, and we have permission to utilize the resource they provide without a need for written consent, but we acknowledge them as a source. Several mitigation strategies were simultaneously tested in parallel, and the findings were reported separately based on treatment. In this trial, the feed additive treatment is AST. The animals were raised from day 0 to 6 weeks in deep litter pens with pine shavings on concrete flooring, four birds in each pen. The animals were reared under two temperature conditions, i.e., thermal neutral (TN) (*n* = 24) at 21–22 °C and 50% RH, and heat stress (HS) (*n* = 36) at 32–35 °C and 42–50% RH. Animals were provided a normal starter feed from 0 to 21 days and a normal finisher feed on 22–42 days with free access to feed and water. Nutritional compositions of the supplemented diets are listed in Table 1. After 14 days, the HS group was further divided into two dietary regimens, i.e., basal diet HS (*n* = 18, treatment 1), and basal diet with 1.33 mg/kg AX supplement (HSAX) (*n* = 18, treatment 2). The light cycle was set at 1:23 dark/light cycles throughout the trial. The broilers were euthanized on day 42, and ileum tissue samples from randomly selected 6 birds of each group were collected at necropsy.

### 2.3. Astaxanthin-Rich Dietary Supplement

The diet was supplemented with P25HB provided by AstaReal^®^, Inc. (Burlington, NJ, USA). PH25B contains 2.5% (*w*/*w*) dried *H. pluvialis*-algae and other components, including modified starch, gum Arabic, mixed tocopherols, L-ascorbyl palmitate, silicon dioxide, xanthan gum, γ-cyclodextrin, polysorbate 80, rosemary extract, and ferulic acid. A comparable nutritional composition of *H. pluvialis*-algae is listed in Appendix A). The natural forms of astaxanthin are comprised of mainly mono-esterified, followed by di-esterified and free forms, i.e., 3,3′ –dihydroxy-β and β-carotene-4, 4′ –dione (C_40_H_52_O_4_ free form). Fuji Health Sciences, Inc. AstaReal ORAC value is reported at 2,822,200 μmol TE/100 g, supported by Non-US Gov’t: Brunswick Laboratories Test Report Batch No. B-10267b-2010.

### 2.4. Growth Performance

Weekly feed intake was recorded, and the average daily feed intake (ADFI), average daily gain ratio (ADG), and feed conversion ratio (FCR) were calculated. The body weight (BW) of each bird was recorded using a Mettler Toledo scale before heat stress treatment and at the end of the heat treatment.

### 2.5. Tissue Sample Collection

Immediately after euthanizing, ileum tissues were collected from randomly selected 6 birds of each group, snapped-frozen in liquid nitrogen, and stored at −80 °C.

### 2.6. Total RNA Extraction and cDNA Preparation

Total RNA was isolated from frozen tissues (50–100 mg) using TRIzol reagent (Invitrogen, Carlsbad, CA, USA) according to the manufacturer’s instructions. The concentration of total RNA was determined using NanoPhotometer^®^ P330 (IMPLEN, Los Angeles, CA, USA). Complementary DNA (cDNA) was synthesized from 1 µg of total RNA (20 µL reaction of RT mixture) using a High-Capacity cDNA Reverse Transcription Kit (Applied Biosystems, Foster City, CA, USA) and further diluted with nuclease-free water (1:25) for the qPCR reaction below.

### 2.7. Bioinformatics: Genome Assembly and Gene Primer Design

The National Center for Biotechnology Information (NCBI) genome browser was used to search and compile genes for *Gallus gallus domesticus* related to heat stress, oxidative stress, cytoprotective, epithelial integrity, transcription factors, and housekeeping genes. The NCBI-Basic Local Alignment Search Tool (BLAST) was used to design primers for polymerase chain reaction (PCR) from the accession numbers obtained from the list of genes (Table 2). The primer parameters were set for a PCR product size between a minimum of 100 and maximum of 250 for 5 primers to return. The primer melting temperatures were set for a minimum of 55 °C, optimum of 57 °C, and maximum of 60 °C with a maximum Tm difference of 3 °C. The exon junction span was set so that the primers must span an exon–exon junction. The organism specified was *Gallus gallus* (taxid 9031). The forward and reverse primer sequences (5′->3′) were then submitted to Integrated DNA Technologies (Coralville, IA, USA) for synthesis.

### 2.8. Quantitative Real-Time RT-PCR (qPCR)

The qPCR was performed using PowerUp SYBR Green Master Mix (Applied Biosystems, Foster City, CA, USA) on a StepOne Plus real-time PCR system (Applied Biosystems, Foster City, CA, USA). The qPCR reaction mixture consisted of 3 µL of cDNA, 5 µL of PowerUp SYBR Green Master Mix, and 1 µL of each forward and reverse primer (5 µmol concentration) to make a final reaction mixture of 10 µL. Specific primer pairs for the detection of each gene were designed using the NCBI Primer-Blast tool (Table 2). The qPCR reaction was carried out following standard cycling mode. The amplification conditions were 50 °C for 2 min (hold), 95 °C for 2 min (hold), followed by 40 repeat cycles of 95 °C for 15 s (denaturation), 60 °C for 15 s (annealing), and 72 °C for 1 min (extension). A melting curve was also generated to confirm SYBR Green-based objective amplicon, and further qPCR products were confirmed using 2% agarose gel electrophoresis. Three housekeeping genes, glyceraldehyde 3-phosphate dehydrogenase (*GAPDH*), beta-actin (*ACTB*), and TATA box-binding protein (*TBP*), were analyzed in triplicate in each bird to determine the most stable housekeeping gene. Based on the uniformity of expression levels across samples, *ACTB* was chosen as the housekeeping gene. Gene expression level was determined using cycle threshold (Ct) values following the standard curve method after normalization with housekeeping genes. Fold change for each gene was calculated using the 2^−ΔΔCt^ method. Data for fold change were presented as mean ± standard error [35].

### 2.9. Gene Ontology

Significantly differentially expressed genes identified from the qPCR procedure were searched in the *Ensembl* genome database for chicken (GRCg6a) species to obtain gene ontologies (GO) information (https://uswest.ensembl.org/Gallus_gallus/Info/Index, accessed on 4 April 2024) [36]. The GO included the cellular component, molecular function, and biological process of these genes identified with ENSGAL transcript IDs.

### 2.10. Tissue Histology Using Hematoxylin and Eosin Staining

A section of the ileum was placed in a scintillating vial with 10% neutral buffered formalin (NBF) and stored at ambient temperature in preparation for Hematoxylin and Eosin (H&E) staining slides. Cross-sectional pieces of ileum were also prepared and fixed overnight in 4% paraformaldehyde (PFA), then incubated in a 15% sucrose solution for 3 h, embedded in optimal cutting temperature (OCT) compound, and stored at −80 °C. The ileum tissues were sectioned at 6 μm thickness, stained with H&E, and mounted to slides at the Histology Core Facility at the John A. Burns School of Medicine, University of Hawaii at Manoa. Slides were observed under both 5× and 20× objective lenses, and images were captured on a Leica Thunder microscope (Danaher, Chicago, IL, USA). The criterion for villus selection was based on the presence of intact lamina propria, and fifteen villi were assessed per sample. Villus height (VH) was measured from the tip of the villus to the crypt, crypt depth (CD) from the villus base to the submucosa, villus surface area (VSA) calculated as the villus base width plus the villus apical width divided by two, and the ratio of villus height to crypt depth (VH/CD) [7,8,37]. The measurements were recorded using Image J Fiji software ver 1.54i [38].

### 2.11. Statistical Analysis

The statistical analysis was performed using the Kruskal–Wallis rank sum test with statistical significance set at *p* < 0.05, followed by the Dunn post hoc test for comparison between three groups, i.e., TN, HS, and HSAX, and the *p*-value adjusted using the Bonferroni method. Growth performance measurements were calculated based on the data collected at the end of the 42-day trial period. Analysis was conducted using the R open source program, libraries ‘FSA’, ‘dunn.test’, and ‘gplots’, R Core Team (2023) (https://www.R-project.org/, accessed on 4 April 2024) [39].

## 3. Results

### 3.1. Growth Performance

At the end of the 42-day poultry trial, the TN group growth performance indicators were found to be significantly higher for BW, ADFI, and ADG compared to the HS and HSAX groups by conducting a Kruskal–Wallis test to evaluate differences among the three groups (*p* < 0.05). Although the results clearly showed negative impacts of heat on the HS and HSAX groups, when performing the post hoc pairwise comparisons using Dunn’s test and applying the Bonferroni correction to control increased risk of error due to multiple comparisons, the negative impacts were found to be more significant in the HS group compared to the TN (*p* < 0.01), whereas the negative impact was slightly less in the HSAX group (*p* < 0.05). In addition, the FCR showed the TN group was significantly lower in feed conversion, requiring less feed to maintain body weight compared to the HS (*p* < 0.05). (Table 3, Figure 1).

### 3.2. Quantitative Real-Time RT-PCR (qPCR) Gene Expression

For gene expression studies, three genes were considered for housekeeping genes, i.e., *GAPDH*, *ACTB*, and *TBP*. *ACTB* was selected for its high and relatively stable expression under experimental conditions performed, making it a reliable reference for normalizing the gene expression data collected. The Kruskal-Wallis method for analyzing statistical significance of differentially expressed genes was conducted using the mean fold change values, plus or minus standard deviation at *p* < 0.05 (Table 4). The gene expression heat map was also visualized by the comparative differences in fold change (Figure 2). A more detailed analysis of the cytoprotective capacity genes (Figure 3), and epithelial integrity genes (Figure 4) provides post hoc pairwise comparisons between the groups using the Dunn’s test and Bonferroni correction to determine statistical significance of gene expression. For each of the genes determined as statistically significant, the gene ontology Table 5, provides a descriptive analysis of factors including: cellular component, molecular function and biological processes associated with the molecular mechanisms and functions.

#### 3.2.1. Cytoprotective Capacity Genes

Performing the post hoc pairwise comparisons using Dunn’s test and applying the Bonferroni correction, the studies showed the impact of HS and AOX treatment expressed through the cytoprotective capacity genes, which resulted in the HSAX treatment group upregulating higher than the HS group in *HSF2* (*p* = 0.017), *SOD2* (*p* = 0.002), and *TXN* (*p* = 0.052) (Table 4, Figure 3). The HSAX group was found to be upregulated higher than the TN group in *GPX3* (*p* = 0.033). The results indicate the AOX treatment appears to increase cytoprotective capacity.

#### 3.2.2. Epithelial Integrity Genes

The expression of the epithelial integrity genes study analyzed with the Dunn’s test and applying the Bonferroni correction resulted in the HSAX treatment group upregulating higher than the TN group for *LOX* (*p* = 0.011), and the HSAX group upregulating higher than the HS group for *CLDN1* (*p* = 0.017), and *MUC2* (*p* = 0.003) (Table 4, Figure 4). The results indicate the AST treatment appears to have a positive effect on maintaining epithelial integrity.

### 3.3. Ileum Histomorphology

The studies showed the impact of HS and AST treatment on the ileum histomorphology (Figure 5). The Dunn’s test and applying the Bonferroni correction were applied to analyze statistical comparisons between the groups (Table 6). The VH measurements from the tip of the villus to the junction of villus and crypt showed a marginally significant difference where the TN villi were slightly longer than HSAX (*p* = 0.055). The CD measurements were defined from the villus base to the submucosa and were found to be significantly different comparing all measurements, where HS was smaller than TN (*p* = 3.77 × 10^−2^) and HSAX (*p* = 2.63 × 10^−2^), respectively, and HSAX was larger than the HS (*p* = 9.30 × 10^−7^). The VSA determined as the villus base width plus the villus apical width divided by two was calculated, and HS exceeded HSAX (*p* = 0.004), the only significant group comparison. The overall VH/CD ratio resulted in significant differences between all groups. with the ratios of the HS being the highest compared to TN (*p* = 3.59 × 10^−1^) and HSAX (*p* = 2.41 × 10^−7^), respectively. In addition, the TN group was significantly different higher than the HSAX (*p* = 4.17 × 10^−4^) (Figure 6).

## 4. Discussion

Results showed both treatments under HS experienced reduced growth performance in comparison to the TN group, while the group treated with AST showed a slightly lesser decline. Results further showed the AST treatment group was significantly upregulated in the cytoprotective gene expression for *HSF2*, *SOD2*, *GPX3*, and *TXN*, as well as the upregulation of epithelial integrity genes *LOX*, *CLDN1*, and *MUC2*. Elucidating the molecular mechanisms underlying cellular responses to environmental stressors such as heat stress contributes to our understanding of how animals adapt and survive adverse conditions and paves the way for targeted therapeutic strategies aimed at mitigating harmful effects on poultry exposed to environmental challenges.

### 4.1. Growth Performance

Growth performance is anticipated to suffer under any stress condition, and in particular HS [40]. Although the growth performance indexes did not show any significant benefit from an AST supplement based on BW measurements related to feed intake and weight gain, the health of the poultry was further elucidated through the gene expression and tissue histomorphology studies providing insights into mechanisms affecting the health and wellbeing of the broilers.

### 4.2. Gene Ontology Enrichment and Expression Analysis

#### 4.2.1. Cytoprotective Capacity Genes

*HSF2* is noted to be found in the nucleus and cytoplasm, where the molecular function of RNA polymerase II cis-regulatory sequence-specific in DNA-binding transcription factor activity occurs. *HSF2* affects the cellular response through positive regulation of transcription from the RNA polymerase II promoter, thereby regulating the transcription of the DNA template (Table 5) [41]. The results suggest AST treatment has a positive effect on the significant upregulation of *HSF2* in the HSAX over the HS group, indicating an effective response of heat shock transcription factors in response to heat stress.

*SOD2* is located in the mitochondrion and functions as an oxidoreductase enzyme involving manganese ion binding in response to hydrogen peroxide negative regulation of Oxidative stress (OS) and the removal of superoxide radicals (Table 5) [42,43]. The results of HSAX upregulated over HS support AST having a positive effect on AOX production to mitigate free radical damage from heat-induced OS by clearing mitochondrial ROS through the electron transport chain, transforming superoxide into oxygen or hydrogen peroxide.

*GPX3* is located in the extracellular space involved in selenium and identical protein binding in response to hydrogen peroxide catabolic process negative regulation of OS (Table 5) [42,43]. The study results appear to indicate the endogenous capability of producing *GPX3* in response to OS provided in the data, where there is no significant difference between the TN and HS groups. However, the application of AST provides a significant advantage in the upregulation of HSAX over the TN group, serving as a boost to the cellular ability to detoxify hydrogen peroxide.

*TXN* is found in the nucleus, cytosol, and extracellular region, supporting protein disulfide oxidoreductase activity for cell redox homeostasis through the balancing of biomolecules positively and negatively affecting the regulation of transcription by RNA polymerase II and DNA binding (Table 5) [41]. The results of HSAX significantly upregulated over HS are consistent with AST having a positive AOX effect in supporting cell redox homeostasis by reducing oxidized cysteine residues and the cleavage of disulfide bonds to reduce free radical damage from OS.

#### 4.2.2. Epithelial Integrity Genes

*LOX* is found in the extracellular space responsible for collagen fibril organization and binding from peptidyl-lysine oxidase enzyme activity involving copper ion binding (Table 5) [44,45]. The results show AST has a positive correlation with promoting tissue integrity in the significant upregulation of the HSAX over the TN Group, indicating an increase in cross-linking collagen and elastin, a possible enhancement of a repair mechanism.

*CLDN1* is an integral component of the cell membrane, particularly within the bicellular tight junction, and provides structural biomolecular activity for the assembly, adhesion, and regulation of ion transport (Table 5) [44,45]. The results of the study imply that AST has a positive effect in support of membrane integrity through the upregulation of the HSAX over the HS group. AST appears to improve cell-to-cell adhesion, acting as a physical barrier preventing solutes and water from leaking into intercellular spaces between the cells.

*MUC2* is located both intracellular and extracellular of the membrane as a supramolecular fiber for protein, virion, and antigen binding for maintenance of the gastrointestinal epithelium through intestinal cholesterol homeostasis and macrophage activation involving the immune response (Table 5) [44,45]. The results show the upregulation of HSAX over the HS group, indicating supporting the AST treatment may help maintain the integrity of the mucus barrier in the gastrointestinal tract by promoting and maintaining an insoluble mucous barrier for protection of the epithelial surfaces.

### 4.3. Ileum Histomorphology

The ileum is the final section of the small intestine where enzyme molecules may adsorb and attach to the surface area, and the final absorption of nutrients occurs before exiting to the ceca. The wall of the ileum is made up of folds of villi, tiny projections that increase the surface area for adsorption and absorption that transport amino acids and glucose produced through digestion into the hepatic portal vein and the liver.

The VH of the HSAX measured significantly less than TN. This appears to be the result of physiological changes, shortening the villi and altering the epithelial cell and morphology, which may be an adaptive mechanism to preserve intestinal function and optimize nutrient absorption [19,23,38].

There were significant differences in the CD, which correlates to the intestinal stem cells and progenitor cells responsible for replenishing the epithelial lining. Crypts also play a crucial role in maintaining the mucosal barrier function, helping to prevent entry of luminal pathogens, toxins, and antigens. This research study showed a decreased measurement of the HS group compared to the TN, an indicator of the effect of heat on the mucosal barrier protective mechanism. Also, highly significant was the difference in the HSAX group in comparison to the HS group, a potential benefit of the AST to promote a mucosal barrier to serve as protection from undesirable intrusions. The statistical difference in crypt depth between the TN and HS indicates potential damage for epithelial regeneration, mucosal barrier function, and immune regulation in the ileum.

The VSA showed a significant difference where the calculation for HS was significantly greater than HSAX. The villus surface area is an indicator for absorption of nutrients, but it would appear in this case that the heat stress effect is detrimental and results in larger surface areas than normal as compared to the TN, resulting in impaired epithelial integrity and tight junctions. Interestingly, there is no difference between the TN and HSAX groups, which may be attributed to AST supplementation to maintain normal nutrient absorption. The higher measurements in the HS group may be the result of increased permeability of the intestinal epithelium, known as a ‘leaky gut’, and increased susceptibility to negative health impacts for inflammation, immune dysregulation, digestive disorders, nutrient malabsorption, and increased susceptibility to infections.

The overall VH/CD ratio can be used to assess structural integrity and functional status of the ileum, providing valuable insights into the balance between epithelial regeneration and turnover. The study results show disruption and the effects of heat on the intestinal health, where the HS ratio is clearly higher than the TN and HSAX. Interestingly, the HSAX ratio is clearly lower than the TN and HS groups, which possibly reveals insights to the protective mechanisms related to the cytoprotective capacity and epithelial integrity.

## 5. Conclusions

In conclusion, our experimental findings indicate AST, a potent antioxidant, shows promise in mitigating molecular changes in heat-induced oxidative stress in the ileum, suggesting therapeutic potential for promoting cytoprotective capacity and protecting epithelial integrity. Although AST does not provide macronutrient value in the diet and therefore showed no positive improvement to growth performance indicators, its supplement as a feed additive shows promise for protection from cellular damage. The observed histomorphology also indicates AST appears to have an effect on epithelial integrity, specifically in the tight junctions to maintain nutrient absorption in the ileum. This research study integrates current knowledge, providing insights into the molecular mechanisms and potential interventions for protecting epithelial integrity by increasing cytoprotective capacity during thermal stress. Nonetheless, knowledge gaps remain in the understanding of the endogenous cellular mechanisms involved in cellular protection from heat-induced oxidative stress. Further research is needed to elucidate the mechanisms underlying the gene expression and observed histomorphology to assess the effects of heat-induced oxidative stress on intestinal physiology and function.

## Figures and Tables

**Figure 1 animals-14-01932-f001:**
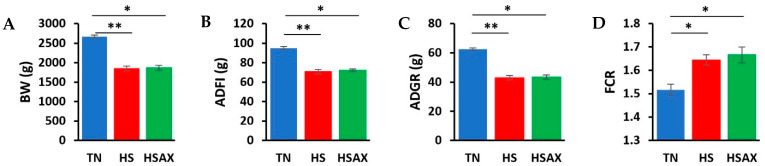
Growth performance indicators of Cobb 500 broilers. Three groups were compared: TN, HS, and HSAX. Dunn post hoc test for pair-wise comparison of statistical significance represented: (**A**) BW (g), TN vs. HS (*p* = 0.010); TN vs. HSAX (*p* = 0.028); (**B**) ADFI (g), TN vs. HS (*p* = 0.005); TN vs. HSAX (*p* = 0.050); (**C**) ADG (g), TN vs. HS (*p* = 0.010); TN vs. HSAX (*p* = 0.0278); and (**D**) FCR, TN vs. HS (*p* = 0.023); TN vs. HSAX (*p* = 0.012). * Differences between treatments significant at *p* < 0.05; ** differences between treatments significant at *p* < 0.01.

**Figure 2 animals-14-01932-f002:**
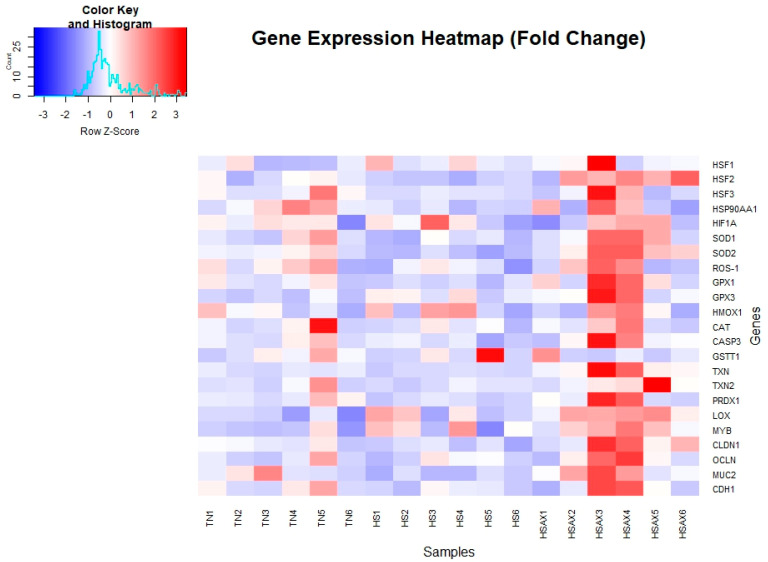
Gene expression fold change heatmap for ileum tissue of Cobb 500 broilers. Three groups were compared: TN, HS, and HSAX. The HSAX group expression values show shades of red with significant higher expression, whereas blue indicates lower levels of expression in the TN and HS groups.

**Figure 3 animals-14-01932-f003:**
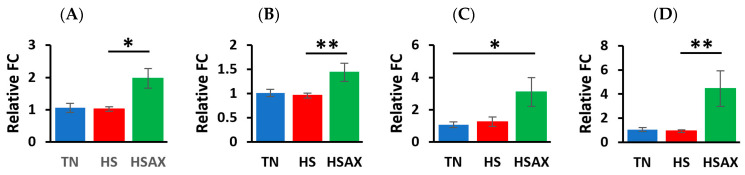
Effects of heat stress and astaxanthin treatment on the cytoprotective capacity gene expression in the ileum tissue of Cobb 500 broilers. Three groups were compared: TN, HS, and HSAX. Dunn post hoc test for pair-wise comparison of statistical significance represented: (**A**) *HSF2*, HS vs. HSAX (*p* = 0.017); (**B**) *SOD2*, HS vs. HSAX (*p* = 0.002); (**C**) *GPX3*, TN vs. HSAX (*p* = 0.033); and (**D**) *TXN*, HS vs. HSAX (*p* = 0.002). * Differences between treatments significant at *p* < 0.05; ** differences between treatments significant at *p* < 0.01.

**Figure 4 animals-14-01932-f004:**
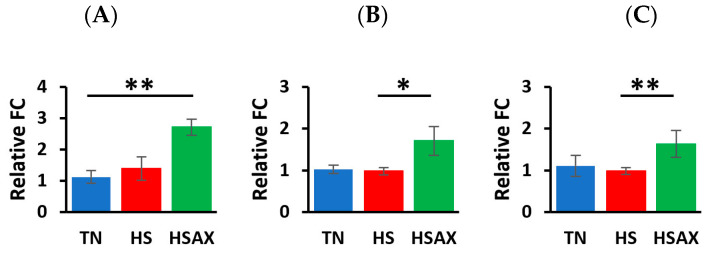
Effects of heat stress and astaxanthin treatment in the epithelial integrity gene expression in ileum tissue of Cobb 500 broilers. Three groups were compared: TN, HS, and HSAX. Dunn post hoc test for pair-wise comparison of statistical significance represented: (**A**) *LOX*, TN vs. HSAX (*p* = 0.011); (**B**) *CLDN1*, HS vs. HSAX (*p* = 0.017); and (**C**) *MUC2*, HS vs. HSAX (*p* = 0.003). * Differences between treatments significant at *p* < 0.05; ** differences between treatments significant at *p* < 0.01.

**Figure 5 animals-14-01932-f005:**
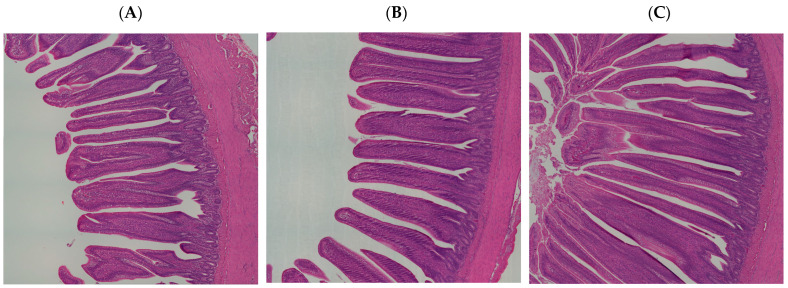
Effect of chronic heat stress on Cobb 500 broiler ileum histomorphology from hematoxylin and eosin (H&E) stained images (scale bar = 200μm). Three groups were measured and compared: (**A**) TN, (**B**) HS, and (**C**) HSAX.

**Figure 6 animals-14-01932-f006:**
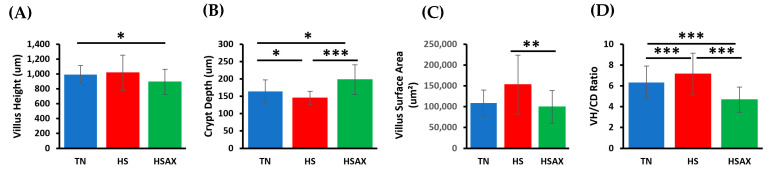
Effects of heat stress and astaxanthin treatment on Cobb 500 broiler ileum tissue morphology. Three groups were compared: TN, HS, and HSAX. Dunn post hoc test for pair-wise comparison of statistical significance represented: (**A**) VH, TN vs. HSAX (*p* = 0.055); (**B**) CD, TN vs. HS (*p* = 3.77 × 10^−2^), TN vs. HSAX (*p* = 2.63 × 10^−2^); HS vs. HSAX (*p* = 9.30 × 10^−7^); (**C**) VSA, HS vs. HSAX (*p* = 0.004); and (**D**) VH:CD, TN vs. HS (*p* = 3.59 × 10^−1^), TN vs. HSAX (*p* = 4.17 × ^10−4^), HS vs. HSAX (*p* = 2.41 × 10^−7^). * Differences between treatments significant at *p* < 0.05; ** differences between treatments significant at *p* < 0.01; *** differences between treatments significant at *p* < 0.001.

**Table 1 animals-14-01932-t001:** Basal diet ingredients and calculated analysis of broiler diets used in the mixture design for thermal neutral and heat stress, including astaxanthin supplement.

Ingredients (%)	Starter	Finisher
Corn	54.86	63.14
Soybean Meal	39.49	29.59
Soybean oil	2.00	4.50
Limestone	1.27	0.85
Monocalcium phosphate	0.75	0.50
L-lysine (98–99%)	0.23	0.18
DL-methionine (99%)	0.14	0.12
L-threonine (98–99%)	0.20	0.16
Sodium chloride	0.43	0.35
Sodium bicarbonate	0.12	0.10
Vitamin–mineral premix ^1^	0.50	0.50
Astaxanthin supplement ^2^	0.01	0.01
Total	100	100
Calculated analysis		
ME (kcal/kg)	2909	3203
Soybean Meal-CP (%)	22.09	18.07
Calcium	0.75	0.52
Total Phosphorus	0.57	0.47
dig Phosphorous	0.30	0.23
L-lysine	1.39	1.10
dig L-lysine	1.25	0.99
DL-methionine	0.48	0.41
dig DL-methionine	0.45	0.39
L-cysteine	0.43	0.38
L-threonine	1.03	0.85
dig L-threonine	0.85	0.69
L-tryptopha	0.33	0.26
DL-methionine + L-cysteine	0.91	0.80
L-arginine	1.61	1.31
L-valine	1.22	1.03
L-isoleucine	0.93	0.76
L-leucine	1.89	1.63
Neutral detergent fiber (% DM)	9.13	8.78
Crude fiber	3.97	3.46
Sodium (mg/kg)	0.22	0.18
Chloride (mg/kg)	0.30	0.25
Choline (mg/kg)	1419	1200
Astaxanthin (mg/kg)	---	1.33

^1^ Vitamin–mineral premix (per kg of diet): vitamin A (trans-retinyl acetate), 10,000 IU; vitamin D3 (cholecalciferol), 3000 IU; vitamin E (all-rac-tocopherolacetate), 30 mg; vitamin B1, 2 mg; vitamin B2, 8 mg; vitamin B6, 4 mg; vitamin B12 (cyanocobalamin), 0.025 mg; vitamin K3 (bisulphatemenadione complex), 3 mg; choline (choline chloride), 250 mg; nicotinic acid, 60 mg; pantothenic acid (D-calcium pantothenate), 15 mg; folic acid, 1.5 mg; betaine anhydrous, 80 mg; D-biotin, 0.15 mg; zinc (ZnO), 80 mg; manganese (MnO), 70 mg; iron (FeCO_3_), 60 mg; copper (CuSO_4_·5H_2_O), 8 mg; iodine (KI), 2 mg; selenium (Na_2_SeO_3_), 0.2 mg. ^2^ Astaxanthin was mixed with the soybean oil and supplemented in the diet during feed mixing.

**Table 2 animals-14-01932-t002:** *Gallus gallus* oligonucleotide primers used for real-time RT-PCR analysis.

Gene	NCBI Accession No.	Primer Set (5′-3′)
*GAPDH*	X00182	F: 5′-CGCAAGGGCTAGGACGG
		R: 3′-GCGCTCTTGCGGGTACC
*ACTB*	NM_205518.1	F: 5′-GAGAAATTGTGCGTGACATCA
		R: 3′-CCTGAACCTCTCATTGCCA
*TBP*	NM_205103.1	F: 5′-TAGCCCGATGATGCCGTAT
		R: 3′-GTTCCCTGTGTCGCTTGC
*HSF1*	NM_001305256.1	F: 5′-AAGGAGGTGCTCCCAAAGTA
		R: 3′-TTCTTTATGCTGGACACGCTG
*HSF2*	NM_001167764.2	F: 5′-TCTTTTTACAAGCTCCGTGC
		R: 3′-TCCCTTTGTCTCCATTTTGGT
*HSF3*	NM_001305041.1	F: 5′-TTCAGCGATGTGTTTAACCCT
		R: 3′-GGAGGTCTTTTGGATCCTCT
*HSP90AA1*	NM_001109785.1	F: 5′-GATAACGGTGAACCTTTGGG
		R: 3′-GGGTAGCCAATGAACTGAGA
*HIF1A*	XM_015287264.4	F: 5′-GTCACCGACAAGAAGAGGAT
		R: 3′-GTCTCTAGCTCACCAGCATC
*SOD1*	NM_205064.1	F: 5′-CAACACAAATGGGTGTACCA
		R: 3′-CTCCCTTTGCAGTCACATTG
*SOD2*	NM_204211.1	F: 5′-CCTTCGCAAACTTCAAGGAG
		R: 3′-AGCAATGGAATGAGACCTGT
*ROS-1*	NM_205257.2	F: 5′-AAACTACAGCTGGTGTTCCC
		R: 3′-CTAAGTTCTCGGCCTTCCAT
*GPX1*	NM_001277853.2	F: 5′-AATTCGGGCACCAGGAGAA
		R: 3′-CTCGAACATGGTGAAGTTGG
*GPX3*	NM_001163232.3	F: 5′-AATTCGGGCACCAGGAGAA
		R: 3′-CTCGAACATGGTGAAGTTGG
*HMOX1*	NM_205344.2	F: 5′-AATCGCATGAAAACAGTCCA
		R: 3′-CACATGGCAAATAAGCCCAC
*CAT*	NM_001031215.2	F: 5′-TGGCCAATTATCAGAGGGAC
		R: 3′-CTCGCACCTGAGACACATTA
*CASP3*	NM_204725.1	F: 5′-GGTGGAGGTGGAGGAGC
		R: 3′-TGAGCGTGGTCCATCTTTTA
*GSTT1*	NM_205365.1	F: 5′-AACATCCCGTTCGAGTTCAA
		R: 3′-CACTATTTGATGGCCCTGTG
*TXN*	NM_205453.1	F: 5′-GGCAATCTGGCTGATTTTGA
		R: 3′-ACCATGTGGCAGAGAAATCA
*TXN2*	NM_001031410.1	F: 5′-CGATTGAGTACGAGGTGTCA
		R: 3′-CAGAAGAAAACCCCACAAACTT
*PRDX1*	NM_001271932.1	F: 5′-GGTATTGCATACAGGGGTCT
		R: 3′-AGGGTCTCATCAACAGAACG
*LOX*	NM_205481.2	F: 5′-TACTTCCAGTACGGTCTGCC
		R: 3′-CTCTAACATCCGCCCGATAA
*MYB*	NM_205306.1	F: 5′-AGCATATACAGCAGCGATGA
		R: 3′-TTTCTCATCCTCTTCACGGG
*CLDN1*	NM_001013611.2	F: 5′-CATCACTTCTCCTTCGTCAGC
		R: 3′-GCACAAAGATCTCCCAGGTC
*OCLN*	NM_205128.1	F: 5′-CTACAAGCAGGAGTTCGACA
		R: 3′-CTCTGCCACATCCTGGTATT
*MUC2*	NM_001318434.1	F: 5′-TACAGGGAGTTCTCTGTCCA
		R: 3′-TAGGGTGTCTTGACAATCCG
*CDH1*	NM_001039258.2	F: 5′-GAACTTCATCGACGAGAACC
		R: 3′-CGTTGAGGTAGTCGTAGTCC

**Table 3 animals-14-01932-t003:** Growth performance indicators of Cobb 500 broiler chickens.

Measurements	TN	HS	HSAX	*p*-Value
BW (g)	2673.68 ^b^	1848.85 ^a^	1867.83 ^a^	0.005
	±35.71	±61.56	±60.82	
ADFI (g)	94.98 ^b^	70.68 ^a^	72.24 ^a^	0.004
	±1.45	±2.38	±1.14	
ADG (g)	62.65 ^b^	42.99 ^a^	43.45 ^a^	0.005
	±0.86	±1.46	±1.44	
FCR	1.52 ^b^	1.64 ^a^	1.67 ^a^	0.005
	±0.02	±0.02	±0.03	

Letters a and b describe significant differences between treatments at *p* < 0.05.

**Table 4 animals-14-01932-t004:** Genes exhibiting significant differential expression for the ileum tissue of Cobb 500 broiler chickens with *ACTB* as the housekeeping gene for normalization.

Gene	TN	HS	HSAX	*p*-Value
*HSF1*	1.52	2.04	3.72	0.150
	±0.55	±0.61	±1.49	
*HSF2*	1.06 ^a^	1.02 ^b^	1.97 ^a^	0.022
	±0.14	±0.07	±0.31	
*HSF3*	1.09	1.03	1.18	0.220
	±0.23	±0.03	±0.42	
*HSP90AA1*	1.09	1.03	0.93	0.130
	±0.20	±0.06	±0.32	
*HIF1A*	1.04	1.07	1.08	0.895
	±0.12	±0.17	±0.17	
*SOD1*	1.10	1.01	1.59	0.199
	±0.23	±0.13	±0.34	
*SOD2*	1.01 ^a^	0.96 ^b^	1.44 ^a^	0.003
	±0.07	±0.05	±0.19	
*ROS-1*	1.08	0.99	1.17	0.372
	±0.17	±0.13	±0.27	
*GPX1*	1.06	1.00	1.85	0.075
	±0.15	±0.07	±0.44	
*GPX3*	1.07 ^b^	1.26 ^a^	3.10 ^a^	0.027
	±0.18	±0.29	±0.90	
*HMOX1*	1.03	1.10	1.17	0.849
	±0.12	±0.19	±0.21	
*CAT*	1.23	1.14	1.17	0.834
	±0.43	±0.14	±0.29	
*CASP3*	1.06	1.04	1.58	0.130
	±0.16	±0.12	±0.43	
*GSTTI*	1.42	1.25	1.05	0.523
	±0.47	±1.07	±0.62	
*TXN*	1.07 ^a^	0.93 ^b^	4.45 ^a^	0.002
	±0.16	±0.10	±1.46	
*TXN2*	1.82	1.65	3.96	0.103
	±0.97	±0.21	±1.73	
*PRDX1*	1.07	0.97	1.64	0.052
	±0.19	±0.07	±0.54	
*LOX*	1.12 ^b^	1.39 ^a^	2.72 ^a^	0.013
	±0.20	±0.37	±0.26	
*MYB*	1.06	1.20	1.95	0.059
	±0.16	±0.31	±0.23	
*IL-4*	1.57	1.36	1.83	0.117
	±0.73	±0.11	±0.78	
*CLDN1*	1.03 ^a^	0.98 ^b^	1.71 ^a^	0.021
	±0.10	±0.09	±0.35	
*OCLN*	1.13	1.03	1.89	0.368
	±0.29	±0.19	±0.54	
*MUC2*	1.11 ^a^	0.99 ^b^	1.64 ^a^	0.004
	±0.25	±0.09	±0.33	
*CDH1*	1.06	1.01	1.30	0.567
	±0.17	±0.09	±0.37	

Letters a and b describe significant differences between treatments at *p* < 0.05.

**Table 5 animals-14-01932-t005:** Gene ontology of differentially expressed genes in the ileum of 6-week Cobb 500 broilers exposed to 21 days of heat stress and astaxanthin treatment (Ensembl) [36].

Gene	Cellular Component	Molecular Function	Biological Process	Transcript IDs
*HSF2*	nucleus and cytoplasm	DNA-binding transcription factor activity, DNA-binding transcription activator activity, RNA polymerase II-specific, and RNA polymerase II cis-regulatory region sequence-specific DNA binding	cellular response to heat, positive regulation of transcription from RNA polymerase II promoter in response to heat stress, regulation of transcription, DNA-templated	ENST00000368455 and ENST00000452194
*SOD2*	mitochondrion	superoxide dismutase activity, oxidoreductase activity, manganese ion binding, metal ion binding, and identical protein binding	response to oxidative stress, oxidation–reduction process, negative regulation of oxidative stress-induced intrinsic apoptotic signaling pathway, response to hydrogen peroxide, and removal of superoxide radicals	ENSGALT00000019062
*GPX3*	extracellular space	glutathione peroxidase activity, selenium binding, and identical protein binding	response to oxidative stress and hydrogen peroxide catabolic process	ENSGALG00010016480
*TXN*	nucleus, cytosol, and extracellular region	protein disulfide oxidoreductase activity	oxidation–reduction process, cell redox homeostasis, negative regulation of hydrogen peroxide-induced cell death, positive regulation of peptidyl-serine phosphorylation, positive regulation of peptidyl-cysteine S-nitrosylation, negative regulation of transcription by RNA polymerase II, and positive regulation of DNA binding	ENSGALT00000025326
*LOX*	extracellular space	protein-lysine 6-oxidase activity, copper ion binding, and collagen binding	peptidyl-lysine oxidation and collagen fibril organization	https://www.ncbi.nlm.nih. gov/gene/396474 (accessed on 4 April 2024)
*CLDN1*	cell junction, bicellular tight junction, integral component of membrane, and cytoplasm	structural molecule activity	bicellular tight junction assembly, cell adhesion, and regulation of ion transport	ENSGALT00000077095 and ENSGALT00000030190
*MUC2*	extracellular matrix, intracellular membrane-bounded organelle, and supramolecular fiber	protein binding, virion binding, and antigen binding	cholesterol homeostasis, intestinal cholesterol absorption, maintenance of gastrointestinal epithelium, negative regulation of cell growth, and macrophage activation involved in immune response	https://www.ncbi.nlm.nih. gov/gene/414878 (accessed on 4 April 2024)

**Table 6 animals-14-01932-t006:** Ileum histomorphology and statistical significance of Cobb 500 broilers villus and crypt measurements. Measurements for VH, CD, VSA, and VH/CD ratio were obtained in μm units.

Measurements	TN	HS	HSAX	*p*-Value
VH (μm)	992.79 ^b^	1061.89 ^a^	893.08 ^a^	0.047
	±118.97	±235.73	±170.00	
CD (μm)	164.28 ^c^	145.08 ^b^	197.87 ^a^	2.05 × 10^−6^
	±32.67	±19.09	±42.86	
VSA (μm)	108,602.30 ^a^	152,946.60 ^b^	99,497.84 ^a^	0.005
	±31,508.84	±70,507.57	±39,296.00	
VH/CD (μm^2^)	6.31 ^c^	7.13 ^b^	4.67 ^a^	2.39 × 10^−7^
	±1.59	±2.00	±1.21	

Letters a, b, and c describe significant differences between treatments at *p* < 0.05.

## Data Availability

The original data presented in this study are openly available on Github at (https://github.com/sweetiek/Broiler_ileum_astaxanthin; accessed on 26 May 2024).

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
