# Peer review of "Use of Microalgae-Derived Astaxanthin to Improve Cytoprotective Capacity in the Ileum of Heat-Induced Oxidative Stressed Broilers"

_animals, 2024, doi:10.3390/ani14131932_

Round 1

Reviewer 1 Report

Comments and Suggestions for Authors

Comments for animals-3056691

The authors tried to investigate the alleviated effects of microalgae-derived astaxanthin on ileum of chicken broilers induced by heat-induced oxidative stressed, found that microalgae-derived astaxanthin is a potential therapeutic role on broilers’ ileum by enhancing the cytoprotective capacity.

1.      In the section of Simple Summary, the sentences must be simplified according to the research content of the entire manuscript, and reinforce the logical relationship between instead of sentence stacking. In line 22, what is the “both treatment”? It should be written clearly, and it represent the “21-22 °C and 32-35 °C?

2.      It is first emerging in the text, adding the abbreviation of “ROS” for reactive oxygen species, and also including others, carefully check. For example, the abbreviation of catalase in line 72 is the CAT, the Thioredoxin in line 74 is Trx but there is the TXN in the line 282 and table 5, what is difference between two lines? What is full name of LOX in the line 89? The word of caeca in line 51 is same to the ceca in line 87, there should be unified. What is the TN, RH, and HS in line 136? The Haematococcus pluvialis-algae in line 148 should be added the H. pluvialis-algae (the seem to the line 151). There is a “O” in line 154 not “0”. Others.

3.      What is the cytoprotective capacity in line 68?

4.      In the section of introduction, the order of the second paragraph can be moved to the third paragraph. Adding the detail roles of the different regulatory molecules based on the current research manuscript.

5.      In section 2.8, what is the primer sequences for qPCR? It is similar to the table 2 of section 2.7?

6.      In the section of results, the results of tables do not need to be repeated in a bar chart.

7.      Discussion is not a repetition of the results, please discuss based on the research findings.

Comments on the Quality of English Language

Extensive editing of English language required

Reviewer 2 Report

Comments and Suggestions for Authors

Authors have prepared an interesting study. In the current manuscript they should expand on performance data. They must comment more on effects such as feed intake and growth and relate more references on such effects. 

Recent references on the effects of astaxanthin on performance are necessary. Some more recent references may be added in Introduction and discussion part.

The authors have not justified the use of actin as reference gene nor did they optimise it for the current experimental conditions. However, they must expand on the reference and standards of their work.

Also, due to similarity test, several phrases should be re-written and no source of reference should be listed higher than 1%.

Authors must also correct Cobb throughout the text.

Authors must correct conclusions and results to add the main effects of performance.

Chemical analysis of the algae additive and specialised analysis, Total phenolic content, antioxidant potentials should be added.

Reviewer 3 Report

Comments and Suggestions for Authors

Use of Microalgae-Derived Astaxanthin to Improve Cytoprotective Capacity in the Ileum of Heat-Induced Oxidative Stressed Broilers

Dear Authors,

The manuscript is interesting, because describes possibility of application of astaxanthin and its antioxidative effect on cells of ileum during months with high environmental temperature.

It is well prepared, but there are some aspect which must be corrected and explained by Authors. Below I add some suggestions helpful in this process:

Line 66, 69, 73 and 85

Three references per sentence are included in the text in ascending order and must be present respectively: [7-9], [10-12], [15-17] and [20-22].

Line 145

Table 1

In case of soybean meal level of protein in % can be added.

Amino acid also must be described with its percentage content in final product and form or forms of its isomers.

L-Lysine (98-99%) or HCl L-Lysine (78%), DL-Methionine and L-Threonine. The same also in case of chemical composition of diets (calculated analysis) D, L or DL-isomer.

Below the table where information about change number of superscript is required: 2Astaxanthin (please change also this ingredient in ingredient section in Table 3). Second change: 1Vitamin-mineral premix composition must be provide below the Table 3 as in majority of articles (please add also1 after Vitamin-mineral premix in ingredient section of this table).

Line 157

In text of manuscript average daily gain ratio are abbreviated as ADGR, but more appropriate and common form is average daily gain (ADG).

Line 237

The same like in line 157.

Line 250

Significance of differences was determined using Kruskall-Wallis non-parametrical ANOVA (3 decimals are enough), but differences in rows can be also determined in form of letters in a superscripts using Dunn/Bonferroni test. Last 3 columns can be deleted and differences can be presented in treatments columns as letters in superscripts, ie.:

Measurement

TN

HS

HSAX

p-value

BW(g)

2673.68a

± 35.71

1848.85b

± 61.56

1867.83b

± 60.82

0,005

ADFI (g)

94.98a

± 1.45

70.68b

± 72.24 (please check sd, higher than mean value! 1.??)

72.24b

± 1.14

0,004

ADG (g)

FCR (kg/kg)

Letters a, b describe significant differences between treatments at p<0.05.

Description of Table 3 can be shortened to: Growth performance indicators of Cobb-500 broiler chickens. They are described in table and material and methods.

Line 253

Description under the figure can be shortened to form: ‘…Figure 1. Growth performance indicators of Cobb-500 broilers…’ * Differences between treatments significant at p<0.05; ** differences between treatments significant at p<0.01.

Line 270

Table 4

Values for expression of genes must be added in three columns after gene name, before p-value. Table must have the same form like in line 250 with description of significance level under the table.

Line 289, 306 and 340

The same like in line 253 in description of Figure 3.

Line 342

The same like in line 250 in description of Table 3.

Line 349

Only one new reference in Discussion section! Maybe better is to resign from several references in Introduction (34) and move them to Discussion section, or even add more new references to Discussion section. One reference is a precedent from scientific point of view.

Line 371

In text of manuscript is …[20,21,22,36], must be [20-22,36].

Line 470

Appendix A

Maybe it is possible to present in chemical composition of Haematococcus algae instead of:

·       protein – crude protein (CP in g or %),

·       fat – ether extract (EE in g or %)

·       carbohydrates – crude fibre and nitrogen free-extract (respectively CF and NFE in g of %)

Line 472

References

Abbreviated form of Journal’s name required.

Without volumens and page abbreviations in single reference, according to Instructions for authors ( https://www.mdpi.com/journal/animals/instructions), all Authors must be specified in case of each publication.

Ie. Reference no.4:

4. Kers, J.G.; Velkers, F.C.; Fischer, E.A.J.; Hermes, G.D.A.; Stegeman, J.A.; Smidt, H. Host and environmental factors affecting the intestinal microbiota in chicken. Front. Microbiol. 2018, 9, doi:10.3389/fmicb.2018.00235

Round 2

Reviewer 1 Report

Comments and Suggestions for Authors

The author has made careful revisions and can be further published in the journal of animals.

Author Response

Comments 1: The author has made careful revisions and can be further published in the journal of animals.

Response 1: Thank you for your insights and constructive suggestions to improve our manuscript submission. We really appreciate your time and consideration and have implemented all of your recommendations.

Reviewer 3 Report

Comments and Suggestions for Authors

Dear Authors,

Thank you for revision process.

Only 3 suggestions this time:

Line 51

In text of manuscript is proventriculu, must be proventriculus.

Line 167

Percentages are important only for L-lysine, DL-methionine and L-Threonine (only those 3 ingredients) and content of amino acid in final product. In case of monocalcium phosphate, sodium chloride and sodium bicarbonate there is no need to determine it.

The same in case of calculated analysis, without percentage vlues.

Line 283

Significance level in table for different genes must be also specified precisely

Gene

TN

HS

HSAX

p-value

SOD2

1,01 b

0,96b

1,44a

0.003

GPX3

1.07b

1.26b

3.10a

0.027

TXN

1.07b

0.93b

4.45a

0.002

LOX

1.12b

1.39b

2.72a

0.013

CLDN1

1.03b

0.98b

1.71a

0.021

MUC2

1.11b

0.99b

1,64a

0.004

HSF3 in Table 4, in case of p-value 0 can be added on the end (0.220).

The font in references must be uniform.

Best regards
